# Peptide-based quorum sensing systems in *Paenibacillus polymyxa*

Maya Voichek[1] , Sandra Maaß[2] , Tobias Kroniger[2], Dörte Becher[2], Rotem Sorek[1]

***Paenibacillus polymyxa* is an agriculturally important plant growth–promoting rhizobacterium. Many *Paenibacillus* species are known to be engaged in complex bacteria–bacteria and bacteria–host interactions, which in other species were shown to necessitate quorum sensing communication. However, to date, no quorum sensing systems have been described in *Paenibacillus*. Here, we show that the type strain *P. polymyxa* ATCC 842 encodes at least 16 peptide-based communication systems. Each of these systems is comprised of a pro-peptide that is secreted to the growth medium and processed to generate a mature short peptide. Each peptide has a cognate intracellular receptor of the RRNPP family, and we show that external addition of *P. polymyxa* communication peptides leads to reprogramming of the transcriptional response. We found that these quorum sensing systems are conserved across hundreds of species belonging to the *Paenibacillaceae* family, with some species encoding more than 25 different peptide-receptor pairs, representing a record number of quorum sensing systems encoded in a single genome.**

## Introduction

*Paenibacillaceae* is a diverse family of bacteria, many of which are important in agricultural and clinical settings. These include *Paenibacillus larvae*, a pathogen causing the lethal American Foulbrood disease in honeybees (1), as well as *Paenibacillus dendritiformis* and *Paenibacillus vortex*, which are used as models for complex colony pattern formation (2, 3). Arguably, the best studied member of *Paenibacillaceae* is *Paenibacillus polymyxa*, known for its plant growth–promoting traits (4, 5, 6). *P. polymyxa* produces a plethora of beneficial compounds, including the antibiotic polymyxin (7) and the phytohormone indole-3-acetic acid (IAA) (8), and was shown to protect multiple plant species against pathogens (reviewed in reference 9). Because of its beneficial traits, *P. polymyxa* is commercially used as a soil supplement to improve crop growth (10, 11).

Many of the characteristic features associated with the above-mentioned behaviors—production and secretion of antimicrobials, expression of virulence factors, and forming complex colony structures—often require some form of intercellular communication such as quorum sensing (12). Bacteria use quorum sensing to coordinate gene expression patterns on a population level. For example, the quorum sensing network of *Pseudomonas aeruginosa* controls the production of secreted toxins as well as the production of rhamnolipids, which are important for its biofilm architecture (13, 14). Quorum sensing regulates the production of bacteriocin antimicrobials in *Lactobacilli* (15) and *Streptococci* (16, 17), and in *Bacillus subtilis*, quorum sensing systems play a major role in sporulation, biofilm formation, and genetic competence (18). Although *Paenibacillus* bacteria are engaged in similar behaviors, no quorum sensing was reported in any *Paenibacillus* species to date.

Gram-positive bacteria frequently use short peptides, termed autoinducer peptides, as their quorum sensing agents (12). A widespread group of such peptide-based communication systems involves intracellular peptide sensors (receptors) of the RRNPP family, together with their cognate peptides (19). Peptides of these quorum sensing systems are secreted from the cell as pro-peptides and are further processed by extracellular proteases (20) to produce a mature short peptide, usually 5–10 aa long (19, 21). The mature peptides are re-internalized into the cells via the oligopeptide permease transporter (22, 23) and bind their intracellular receptors (24). These receptors act either as transcription factors or as phosphatases, and peptide binding activates or represses their function eventually leading to modulation of the bacterial transcriptional program. For example, in *B. subtilis*, the Rap receptors function as phosphatases that regulate the major transcription factor Spo0A; upon binding to their cognate Phr peptides, the phosphatase activity of Rap receptors is inhibited, leading to activation of Spo0A that alters the transcriptional program (25, 26). In *Bacillus thuringiensis*, the PlcR receptor is a transcription factor that becomes activated when bound to its cognate PapR peptide, leading to expression of virulence factors that facilitate infection of its insect larval host (27). Such peptide-based communication was recently shown also to control the lysis-lysogeny decision of phages infecting many species of *Bacillus* (28, 29).

[1]Department of Molecular Genetics, Weizmann Institute of Science, Rehovot, Israel    [2]Department of Microbial Proteomics, Institute of Microbiology, Center for Functional Genomics of Microbes, University of Greifswald, Greifswald, Germany

Correspondence: rotem.sorek@weizmann.ac.il
Maya Voichek's present address is Institute of Molecular Biotechnology of the Austrian Academy of Sciences (IMBA), Vienna Biocenter (VBC), Vienna, Austria

Here, we report the identification of a large group of peptide-based quorum sensing systems in *P. polymyxa*. We found that pro-peptides of these systems are secreted and further processed and that the mature peptides modulate the transcriptional program of *P. polymyxa*. We further show that these quorum sensing systems are conserved throughout the *Paenibacillaceae* family of bacteria, with some bacteria encoding more than 25 different peptide-receptor pairs in a single genome.

## Results

### Identification of RRNPP-like peptide-receptor pairs in *P. polymyxa* ATCC 842

All members of the RRNPP family of intracellular peptide receptors contain a C-terminal tetratricopeptide repeat (TPR) or TPR-like domain, which forms the peptide-binding pocket in the receptor protein (19, 30). To search for such possible quorum sensing systems in *P. polymyxa*, we first scanned all annotated protein-coding genes of the type strain *P. polymyxa* ATCC 842 (31) for TPR domains using TPRpred (32) (see the Materials and Methods section). As TPR domains are found in diverse protein types, we manually analyzed the genomic vicinity of the identified TPR-containing proteins to search for a peptide-encoding gene characteristic of quorum sensing systems. We found two TPR domain genes that were immediately followed by a short open reading frame with a predicted N-terminal hydrophobic helix, typical of a signal peptide that targets the pro-peptide for secretion (33) (Fig 1A). Two-gene operons that encode the receptor followed by its cognate peptide are a hallmark of many known members of the RRNPP family of quorum sensing systems (19).

We then used BLAST to search for additional homologs of the two putative TPR domain receptors in the *P. polymyxa* ATCC 842 genome

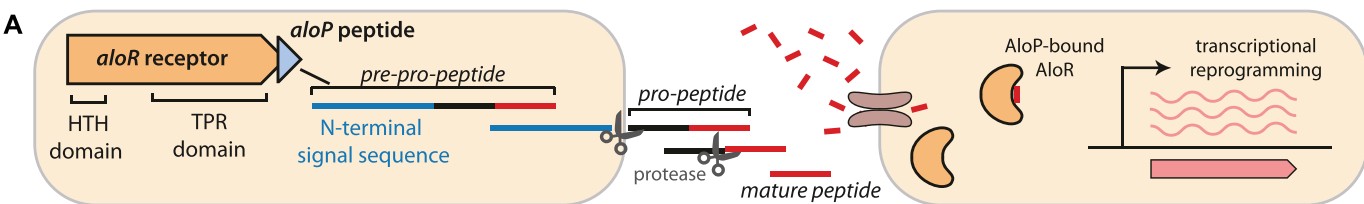

**A**

**B**

| System | AloR locus | AloP small ORF | AloP small ORF sequence |
|--------|-----------|----------------|------------------------|
| Alo1 | PPTDRAFT_00471 | AloP1 (32 aa) | MLRKIGLLIVASSFLVFMTVPGIQILVQVGGS |
| Alo2 | PPTDRAFT_00474 | AloP2 (40 aa) | MLKKTVLLIIAASFLLFIVTVPDQAFNHHSIQPQVTIGGA |
| Alo3 | PPTDRAFT_01046 | AloP3 (41 aa) | MIKKMAFCLAVTSFLLVFTVPAQTANLSHGTIQVQGHIGGS |
| Alo4 | PPTDRAFT_01711 | AloP4 (41 aa) | MKRKIILTILVTSFMLLTLVNPSAGYQNSEHNYEPAQHGEM |
| Alo5 | PPTDRAFT_01713 | (*) | |
| Alo6 | PPTDRAFT_01722 | AloP6 (49 aa) | MKRLIKGLFATTFLLVFIASTQTINLQDFGMPQAFSNGGFSVFGWHIGS |
| Alo7 | PPTDRAFT_02243 | AloP7 (35 aa) | MKKLISRPLSLSLFSLLVFVTSHDLITVFYHGANH |
| Alo8 | PPTDRAFT_02798 | AloP8 (36 aa) | MLKKMALLIVASSFLFFITGPVLTQHIQILVQIGGA |
| Alo9 | PPTDRAFT_02818 | AloP9 (40 aa) | MFKKVVLLIIAASFLLIVTVPDQAFNHHHSIQPNVTIGGF |
| Alo10 | PPTDRAFT_03195 | AloP10α (40 aa) AloP10β (40 aa) | MLRKMALLIVATSFLFIVTLPVQTDGQHQNIVLYDQHGGA MLKKTVMLIIASSFLFTLAVPIYVDGHIGGFQVFATHGGA |
| Alo11 | PPTDRAFT_04223 | AloP11 (43 aa) | MKKKMVFFLAVSFGLMFVASAQLPFEQHQASGLISTQGYIGGA |
| Alo12 | PPTDRAFT_04756 | (*) | |
| Alo13 | PPTDRAFT_04976 | AloP13 (44 aa) | MKKVFLSLLLSLIVVSNIGNLSYFHNITIGNGSQITVSSHGRGG |
| Alo14 | PPTDRAFT_04977 | (*) | |
| Alo15 | PPTDRAFT_05290 | AloP15 (43 aa) | MKKALSSITMVLAFLSVALPIGNTLGDLDSVHPSNHGVFSFTA |
| Alo16 | PPTDRAFT_05358 | AloP16 (50 aa) | MIKKIVFSLLAFNVFILLNFPPVTPPIEDPSDVHSHGLGGWALSSPEPSA |

**Figure 1. Alo systems identified in *P. polymyxa*.**
**(A)** Schematic representation of *aloR-aloP* operon organization and putative peptide processing. HTH, helix-turn-helix; TPR, tetratricopeptide repeat. **(B)** Alo loci identified in *P. polymyxa* ATCC 842. For the *aloR* receptor genes, the locus tag in the Integrated Microbial Genomes (IMG) database (49) is specified. Blue sequence in AloP peptides represents the predicted N-terminal signal sequence for secretion, as predicted by Phobius (50) (see the Materials and Methods section). Asterisks (*) mark cases in which the *aloP* gene was absent or found as a degenerate sequence.

(see the Materials and Methods section). This search revealed 14 additional homologous genes that were not initially recognized by TPRpred, in most cases because their C-terminal TPR domains were divergent and did not pass the default threshold of the TPRpred software (Supplemental Data 1). We found that all but three of the homologs had a short ORF (32–50 aa) encoded immediately downstream to them. We denote these putative communication systems as Alo (Autoinducer peptide Locus), with predicted Alo receptor genes designated *aloR* and the cognate peptide-encoded gene *aloP*. We number Alo loci in the *P. polymyxa* ATCC 842 genome from *alo1* to *alo16* (Fig 1B).

## Alo systems are ubiquitous in *P. polymyxa* strains

Examining the genomes of 13 additional *P. polymyxa* strains using sequence homology searches revealed that the predicted Alo systems are conserved in the *P. polymyxa* lineage (Fig 2A; see the Materials and Methods section). Some of the Alo receptors (e.g., AloR3, AloR4, AloR7, AloR13, AloR15, and AloR16) are highly conserved and appeared in all *P. polymyxa* strains examined. Others appeared variably and were absent from some genomes (Fig 2B). We also found six additional putative peptide-receptor pairs that were not present in *P. polymyxa* ATCC 842 but were detectable in other strains; we denote these as Alo17-22 (Fig 2). Similar observations regarding "conserved" and "variable" quorum sensing systems were reported for peptide-based

systems in other bacteria, for example, in the case of *B. subtilis rap-phr* genes, *rapA-phrA* and *rapC-phrC* are found to be conserved among *B. subtilis* species, whereas other systems, including *rapE-phrE*, *rapI-phrI*, and *rapK-phrK*, occur variably and are often located within mobile genetic elements (34).

In the vast majority of cases, *aloR* genes in the various *P. polymyxa* strains were found just upstream to a peptide-encoding *aloP* gene with an N-terminal hydrophobic helix predicted to target them for secretion (Table S1). Exceptions were *aloR18*, for which we could not find a cognate downstream short ORF, and *aloR5* and *aloR14* in which degenerated peptide sequences were observed. In the Alo12 clade, we could identify a conserved short ORF in most *P. polymyxa* strains (excluding the type strain ATCC 842), yet it lacked the characteristic hydrophobic N-terminal signal sequence marking it for secretion. Orphan receptors, which lack a cognate peptide, have also been observed for RRNPP quorum sensing systems in other bacteria (34, 35).

To understand how common Alo systems are in bacteria, we performed an exhaustive profile-based search for AloR-like proteins in >38,000 bacterial and archaeal genomes (see the Materials and Methods section). This search yielded 1,050 hits in 149 bacteria, essentially all of which belong to the *Paenibacillaceae* family (Table S2). These included species of *Paenibacillus*, *Brevibacillus*, *Saccharibacillus*, *Fontibacillus*, and *Gorillibacterium*. The largest number of Alo receptors found in a single organism was in *Paenibacillus terrae* NRRL B-30644, in which 40 such receptors were

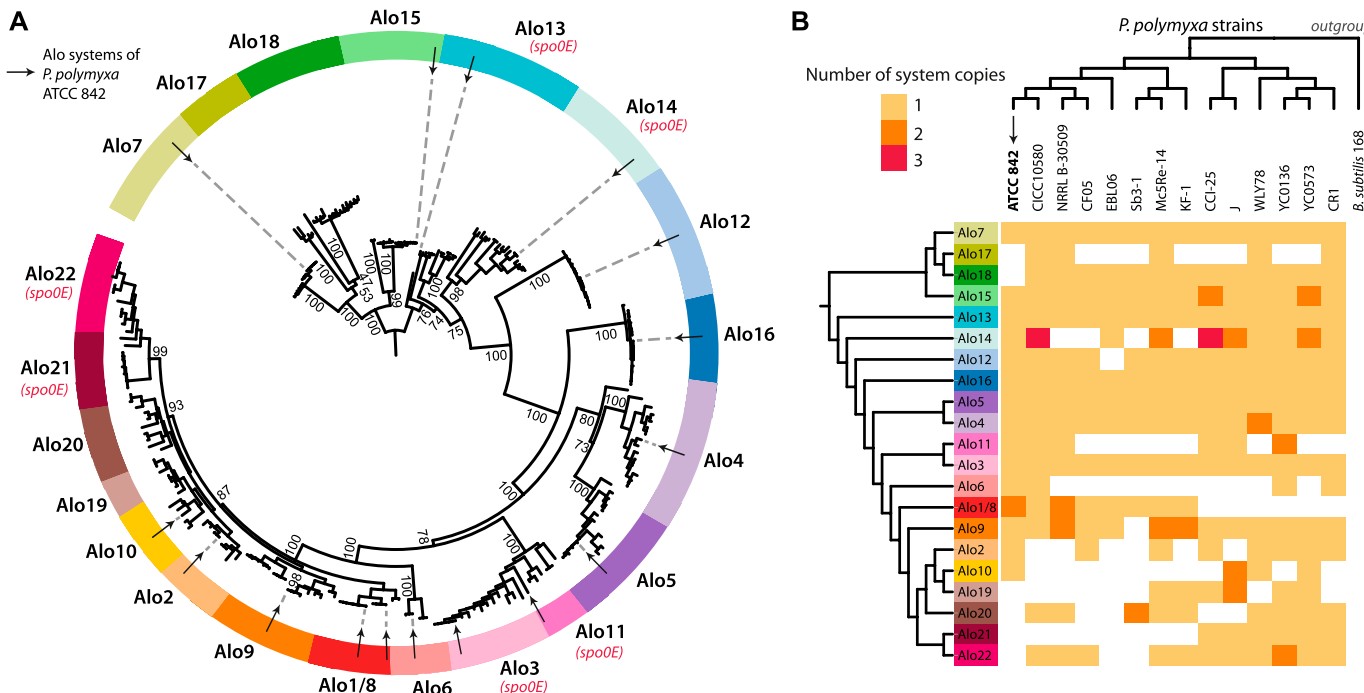

**Figure 2.   Phylogenetic distribution of Alo systems among *P. polymyxa* strains.**
**(A)** Phylogenetic tree of 234 AloR homologs found in 14 *P. polymyxa* strains. Colored segments in the tree circumference define clades of AloR proteins corresponding to the systems found in *P. polymyxa* ATCC 842 (labeled as arrows). Systems immediately followed by a Spo0E-like protein are indicated. Tree was constructed using IQ-Tree (55, 56, 57) with 1,000 iterations, and ultrafast bootstrap support values are presented for major branches. Tree visualization was done with iTOL (58). Alo1/8 refers to a system duplicated in *P. polymyxa* ATCC 842 (where it is represented as Alo1 and Alo8). **(B)** Frequency and distribution of Alo systems in *P. polymyxa* strains. The number of copies per Alo system found within *P. polymyxa* strains is shown using colored squares. The absence of an Alo system in a specific strain is marked by a white square. **(A)** Phylogenetic tree of the Alo systems shown on the vertical axis is derived from the tree in the panel (A). Phylogenetic tree of the strains shown on the horizontal axis is based on the GyrA protein, with GyrA from *B. subtilis* 168 as an out-group.

detected (with cognate peptide-encoding genes detected immediately downstream to 27 of them, Table S3). These results suggest that the Alo system is a widespread quorum sensing system in bacteria belonging to the *Paenibacillaceae* family.

## Mass spectrometry (MS) analysis detects secretion and processing of pro-peptides

Signaling peptides associated with quorum sensing regulators of the RRNPP family are known to be secreted into the growth medium as pro-peptides. The pro-peptides are further processed by extracellular proteases to generate the mature communication peptide (26, 36) (Fig 1A). To determine whether Alo peptides are similarly secreted, we analyzed growth media taken from *P. polymyxa* cultures using MS. For this, we grew *P. polymyxa* ATCC 842 in defined media, removed the bacteria by centrifuging, and took the supernatant (presumably containing the secreted peptides) for MS analysis. Notably, MS was performed without subjecting the peptides to trypsin treatment, to allow detection of peptides in their natural form. We were able to reproducibly detect various fragments of nine out of the 14 Alo peptides predicted in *P. polymyxa* ATCC 842 (Fig 3 and Table S4). In all cases, the fragments included the C terminus of the Alo peptide but not the N terminus, confirming that the N terminus is cleaved during or after the pro-peptide secretion, possibly by a signal peptidase associated with the Sec system (37). As a control, we repeated the same procedure on cultures of *B. subtilis* and found that the *B. subtilis* Phr peptides were also detectable in the MS analysis, with the same

patterns of absent N-termini (Fig 3A and Table S4). In both *B. subtilis* and *P. polymyxa*, MS data indicated multiple possible cleavage sites within the N-terminal signal sequence, as was previously reported also for other Gram-positive bacteria (38) (see the Discussion section). These results show that Alo peptides are secreted to the medium akin to other cases of RRNPP quorum sensing systems.

Following the secretion process, *B. subtilis* Phr peptides are known to be processed further to yield the mature 5-aa peptide. The mature peptide is usually found at the extreme C-terminal end of the pro-peptide, such that it may be released by a single cleavage event (Fig 3A). Such short peptides are challenging to identify using current MS pipelines (39) and indeed our MS analysis of *B. subtilis* conditioned media did not detect the mature 5-aa Phr peptides but rather revealed the Phr pro-peptides with the known mature peptide missing (Fig 3A). The absence of a short peptide from the C terminus of the pro-peptide is therefore a clear signature of the processing event that releases the mature peptide. We were able to detect similar processing events in the *P. polymyxa* peptides as well (Fig 3B). These patterns strongly suggest that some of the AloP proteins generate 6-aa communication peptides, whereas others generate presumably longer mature peptides (Fig 3B).

## Peptide-mediated modulation of the bacterial transcriptional program

All Alo receptors contain a helix-turn-helix DNA-binding domain at their N terminus (Fig 1A), suggesting that they may function as

### A — *B. subtilis* 168 Phr peptides

| Phr | Identified fragment of Phr protein | No. of spectra |
|---|---|---|
| PhrA | MKSKWMSGLLLVAVGFSFTQVMVHA<u>GETANTEGKTFHIA</u>ARNQT | 3 |
| PhrC | MKLKSKLFVICLAAA<u>AIFTAAGVSANAEALDFHVT</u>ERGMT | 6 |
| PhrC | MKLKSKLFVICLAAAA<u>IFTAAG</u>VSANAEALDFHVT<u>ERGMT</u> | 6 |
| PhrF | MKLKSKLLLSCLAL<u>STVFVATTIANAPTHQIEVA</u>QRGMI | 14 |
| PhrF | MKLKSKLLLSCLALS<u>TVFVATTIANAPTHQIEVA</u>QRGMI | 9 |
| PhrG | MKRFLIGAGVAAVILSGWFIAD<u>HQTHSQEMKVA</u>EKMIG | 2 |
| PhrG | MKRFLIGAGVAAVILSGWFIA<u>DHQTHSQEMKVA</u>EKMIG | 1 |
| PhrH | MPIKKKVMMCLAVTLVFGSM<u>SFPTLTNSGGFKES</u>TDRNTTYIDH SPYKLSDQKKALS | 3 |
| PhrH | MPIKKKVMMCLAVTLVFGSMS<u>FPTLTNSGGFKES</u>TDRNTTYIDH SPYKLSDQKKALS | 2 |
| PhrI | MKISRILLAAVILSSVFSIT<u>YLQSDHNTEIKVAA</u>DRVGA | 1 |
| PhrI | MKISRILLAAVIL<u>SSVFSITYLQSDHNT</u>EIKVAA<u>DRVGA</u> | 1 |
| PhrK | MKKLVLCVSIL<u>AVILSGVALTQLSTDSPSNIQVA</u>ERPVGG | 6 |
| PhrK | MKKLVLCVSILAVILSGVALTQL<u>STDSPSNIQVA</u>ERPVGG | 4 |

### B — *P. polymyxa* ATCC 842 AloP peptides

| AloP | Identified fragment of AloP protein | No. of spectra |
|---|---|---|
| AloP11 | MKKKMVFFLAVSFGLMFVASAQLPF<u>EQHQASGLISTQ</u>GYIGGA | 10 |
| AloP11 | MKKKMVFFLAVSFGLMFVA<u>SAQLPFEQHQASGLISTQ</u>GYIGGA | 5 |
| AloP10α | MLRKMALLIVATSF<u>LFIVTLPVQTDGQHQNIVLY</u>DQHGGA | 6 |
| AloP10α | MLRKMALLIVATSF<u>LFIVTLPVQTDGQHQNIVLY</u>DQHGGA | 3 |
| AloP10β | MLKKTVMLIIASSFLF<u>TLAVPIYVDGHIGGFQVF</u>ATHGGA | 3 |
| AloP10β | MLKKTVMLIIASSFLFTLAVPI<u>YVDGHIGGFQVF</u>ATHGGA | 3 |
| AloP13 | MKKVFLSLLLSLIVVSNIGNL<u>SYFHNITIGNGSQITVS</u>SHGRGG | 2 |
| AloP13 | MKKVFLSLLLSLIVVSNIGNLSY<u>FHNITIGNGSQITVS</u>SHGRGG | 1 |
| AloP2 | MLKKTVLLIIAASF<u>LLFIVTVPDQAFNHH</u>SIQPQVTIGGA | 2 |
| AloP2 | MLKKTVLLIIAASF<u>LLFIVTVPDQAFNHHSIQPQ</u>VTIGGA | 1 |
| AloP9 | MFKKVVLLIIAASF<u>LLIVTVPDQAFNHHH</u>SIQPNVTIGGF | 1 |
| AloP9 | MFKKVVLLIIAA<u>SFLLIVTVPDQAFNHHH</u>SIQPNVTIGGF | 1 |
| AloP4 | MKRKIILTILVTSFMLL<u>TLVNPSAGYQNSEH</u>NYEPAQHGEM | 5 |
| AloP4 | MKRKIILTILVTSFMLLTL<u>VNPSAGYQNSEH</u>NYEPAQHGEM | 1 |
| AloP15 | MKKALSSITMVLAFL<u>SVALPIGNTLGDLDSVHPSNHGVFSFTA</u> | 11 |
| AloP15 | MKKALSSITMVLAFL<u>SVALPIGNTLGDLDSVHPS</u>NHGVFSFTA | 8 |
| AloP16 | MIKKIVFSLLAFNVF<u>ILLNFPPVTPPIEDPSDVH</u>SHGLGGWALS SPEPSA | 10 |

*(Predicted 6aa mature peptides — bracket spanning AloP11 through AloP2 rows)*

*(Possibly longer mature peptides — bracket spanning AloP9 through AloP16 rows)*

**Figure 3. Mass spectrometry (MS)–based identification of secreted communication peptides.**
**(A)** Phr peptide fragments detected in the growth media of *B. subtilis* 168. Peptide fragments identified by MS are underlined, with the cumulative number of spectra detected in all tested repeats indicated (see the Materials and Methods section). Known mature communication peptides are in red (26, 45). The two most abundantly identified fragments are presented for each protein by an underline. The mature peptides which were not directly detected are inferred from the C-terminal sequence of the full peptides, presumably after processing. **(B)** AloP peptide fragments detected by MS. Same as (A) for AloP peptide fragments detected in the growth media of *P. polymyxa* ATCC 842.

transcriptional regulators similar to many members of the RRNPP peptide receptors family ([19](ref), [24](ref)). These regulators become activated or repressed once bound by their cognate peptides, leading to alteration of the transcriptional program in response to the quorum sensing peptides. To examine whether Alo peptides affect the transcriptional program of *P. polymyxa*, we selected Alo13, one of the systems conserved across all *P. polymyxa* strains, for further experimental investigation. The sequence of the mature AloP13 peptide, as predicted by the MS analysis, is SHGRGG ([Fig 3B](ref)). Aiming to measure the immediate transcriptional response to the peptide, we incubated early log-phase–growing *P. polymyxa* cells with 5 μM

of synthetic SHGRGG peptide for 10 min. RNA-seq analysis of cells harvested after this short incubation with the peptide identified a set of 10 genes whose expression became significantly reduced when the peptide was added ([Fig 4](ref) and Table S5). This transcriptional response was not observed when a scrambled version (GRGSGH) of the mature AloP13 peptide was added to the medium, indicating that the transcriptional response is specific to the sequence of AloP13 (Table S6).

The most significant reduction in expression was in a gene annotated as Spo0E-like sporulation regulatory protein, whose expression was reduced by 4.5-fold on average in response to the

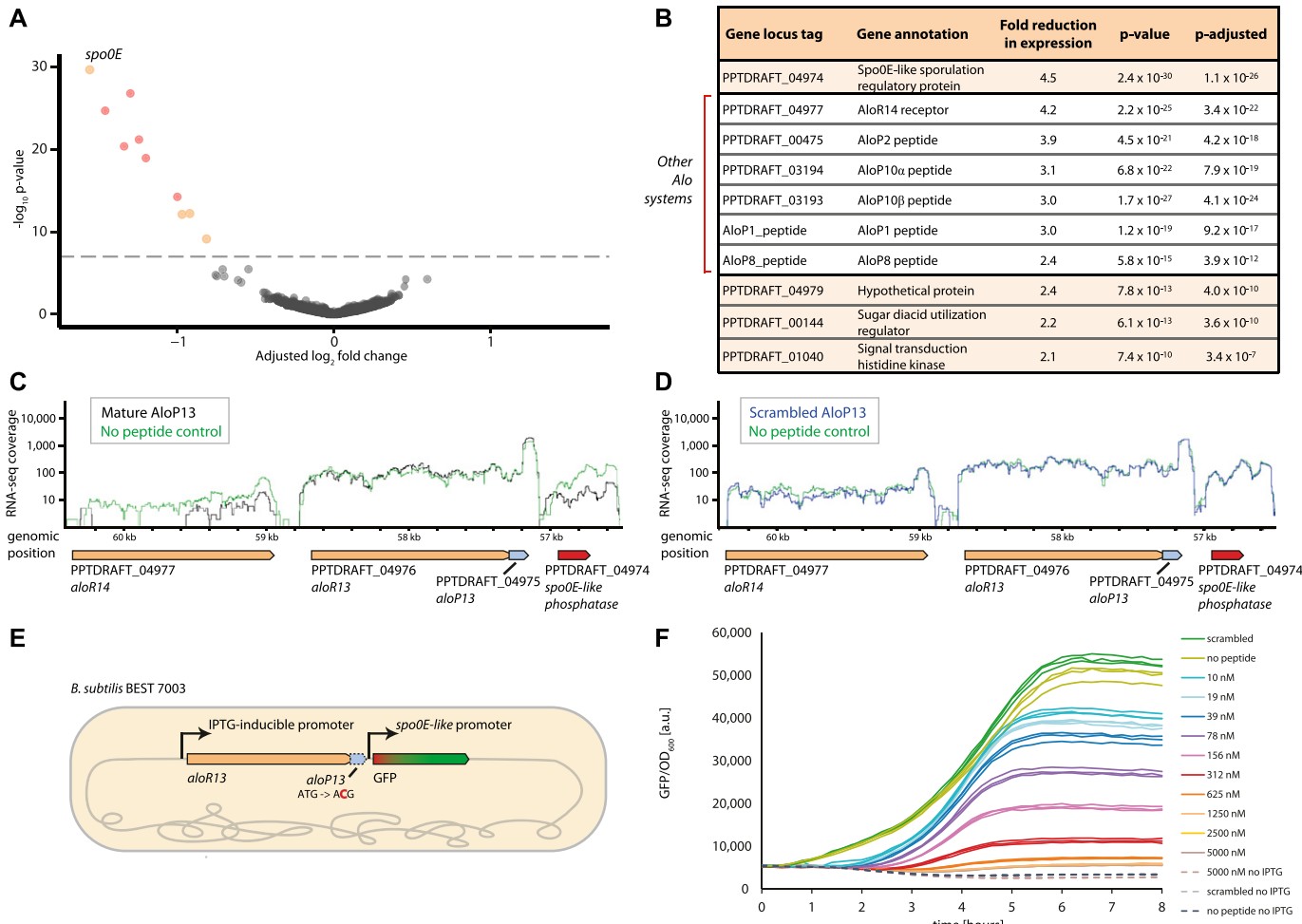

**Figure 4. The mature AloP13 peptide elicits an immediate transcriptional response.**
**(A)** Volcano plot depicting *P. polymyxa* gene expression after incubation with 5 μM of the peptide SHGRGG for 10 min, as compared to control conditions in which no peptide was added. X-axis, $\log_2$ fold expression change (adjusted using the lfcShrink function in the DESeq2 R package ([62](ref)); see the Materials and Methods section). Y-axis, $-\log_{10}$ *P*-value. Average of three independent replicates; each dot represents a single gene. Dots appearing in colors are genes that passed the threshold for statistical significance of differential expression (see the Materials and Methods section). Red dots correspond to genes encoded by Alo systems. **(B)** Differentially expressed genes as appears in panel (A). Fold change represents the average of the three independent replicates. P-adjusted is the *P*-value after correction for multiple hypothesis testing (see the Materials and Methods section). **(C)** RNA-seq coverage of the Alo13-Alo14 locus (Scaffold: PPTDRAFT_AFOX01000049_1.49), in control conditions (green) or 10 min after addition of 5 μM of the SHGRGG peptide (black). RNA-seq coverage is in log scale and was normalized by the number of uniquely mapped reads in each condition. Representative of three independent replicates. **(D)** RNA-seq coverage of the Alo13-Alo14 locus in control conditions (green) or 10 min after addition of 5 μM of a scrambled version (GRGSGH) of the AloP13 peptide. RNA-seq coverage is in log scale and normalized as in the panel (C). Representative of three independent replicates. **(E)** Schematic representation of the genetic construct used in *B. subtilis* to verify the activity of AloR13. The start codon of the *aloP13* gene (dashed blue arrow) was mutated to inactivate it. GFP was placed as a reporter gene instead of the Spo0E ORF. **(F)** GFP fluorescence upon addition of the AloP13 peptide SHGRGG in varying concentrations, or the scrambled mature peptide GRGSGH in 5 μM ("scrambled"), or no peptide added. Dashed lines represent GFP fluorescence measured without the addition of IPTG.

peptide (3.7-, 6-, and 4-fold reduction in the three independent replicates of the experiment) (Fig 4B). This reduction was also verified by reverse transcription quantitative real-time PCR (RT-qPCR) (Fig S1). In *B. subtilis*, Spo0E functions as a phosphatase that regulates Spo0A, the master transcription factor for sporulation, thus inhibiting the initiation of the sporulation pathway (40). Interestingly, the *spo0E*-like gene is encoded directly downstream to the Alo13 locus in *P. polymyxa* (Fig 4C and D). These results imply that the immediate effect of the AloP13 quorum sensing peptide involves down-regulation of the *spo0E*-like gene, which likely leads to a further regulatory cascade at later time points. We note that *spo0E*-like genes occur immediately downstream of 6 Alo systems in *P. polymyxa* (Fig 2A) and downstream to 12 of the *P. terrae* Alo systems (Table S3). This conserved genomic organization suggests that many Alo systems may exert their effect on the cell by modulating the expression of Spo0E-like regulatory phosphatase proteins. Notably, a similar regulatory architecture is observed in the arbitrium RRNPP communication module that regulates phage lysogeny decisions, where the binding of the arbitrium peptide to its AimR receptor also leads to down-regulation of the immediate downstream gene, *aimX* (28).

Most of the additional genes whose expression became reduced following 10 min of exposure to the AloP13 mature peptide were those encoding other AloP peptides, including *aloP1*, *aloP2*, *aloP8*, *aloP10α*, and *aloP10β*, as well as the gene encoding the receptor *aloR14* that is encoded adjacent to the *aloR13* gene (Fig 4). These results suggest that as in other bacteria (35, 41), quorum sensing systems can sometime cross-regulate other such systems in *P. polymyxa*. Combined, these results suggest that the AloP13 mature peptide affects the transcription of specific target genes and verifies Alo systems as quorum sensing systems in *P. polymyxa*.

To verify that the *spo0E* gene is indeed directly controlled by the Alo system, we cloned the entire *alo13* locus into the genome of *B. subtilis*, but replaced the *spo0E* gene with a GFP reporter (Fig 4E). As shown in Fig 4F, activation of the AloR expression results in accumulation of the GFP signal, indicating that AloR positively controls the Spo0E promoter. However, external addition of the mature AloP peptide SHGRGG resulted in reduction of GFP expression, in a manner directly dependent on the peptide concentration, with a near-complete shutoff of GFP expression observed when 5 μM of peptide was added. Expression reduction was observed when as little as 10 nM of SHGRGG peptide was added, which is parallel to activities observed for other RRNPP systems (28). Moreover, addition of the scrambled version of the AloP peptide (GRGSGH) did not affect GFP expression, showing that the effect of the peptide on AloR activity is sequence specific as in other RRNPP systems (35). Overall, these results verify that the Alo13 peptide-receptor system directly controls the expression of the downstream Spo0E protein.

## Discussion

In this work, we identified a large family of putative peptide-based quorum sensing systems in *Paenibacillaceae* bacteria and functionally characterized one of these systems in the type strain *P. polymyxa* ATCC 842. We found that the peptide precursors are secreted to the growth medium and further processed to form putatively mature communication peptides which lead to transcriptional reprogramming. The large number of such systems in a single bacterium (16 in *P. polymyxa* ATCC 842 and at least 27 in *P. terrae*) suggests that *Paenibacillus* may need an unusually large amount of signals to coordinate their complex social traits. Previous studies have predicted that *P. polymyxa* strains may also use AI-2 quorum sensing, based on the presence of components of the AI-2 pathway in its genome, although the production of AI-2 by *P. polymyxa* was not detected (42, 43). In addition, *P. polymyxa* ATCC 842 was found to encode an *agr*-like gene cassette that is homologous to another peptide-based communication system in staphylococci (44). Together with our discovery of the Alo communication systems, to our knowledge, *Paenibacillus* have the largest number of quorum sensing systems reported in any bacteria to date.

Interestingly, none of the Alo systems discovered possess strong sequence similarity to any of the other known RRNPP systems, and hence, they were not identified in previous searches (e.g., see reference 19, and Fig S2). We believe the domain-based analysis strategy that we used to identify TPR-containing genes, rather than direct sequence homology searches, enabled the discovery of additional distant members of the RRNPP family of receptors. It is possible that usage of a similar search strategy on genomes of Gram-positive bacteria may lead to the discovery of additional RRNPP systems.

Previous studies on quorum sensing systems in other bacteria focused on isolating and characterizing the mature signaling peptide. Our MS-based approach identified, for the first time, the pro-peptide remainders that are found in the extracellular medium after secretion, in both *B. subtilis* and *P. polymyxa*. Curiously, the MS spectra suggest that the hydrophobic N-terminal signal sequence can be cleaved in multiple positions, a phenomenon that has been reported in other Gram-positive bacteria such as *Staphylococcus aureus* (38).

In *B. subtilis*, most Phr pro-peptides are processed by a single protease cleavage event, releasing the C-terminal mature peptide. However, in some cases, there are two cleavage events (e.g., PhrH and PhrK), releasing an internal portion of the pro-peptide to form the mature communication peptide (26, 45). Our MS data of *B. subtilis* peptides show that these cases may not be detectable in our analyses, but instead may appear as longer mature peptides (see PhrH and PhrK in Fig 3A). It is therefore conceivable that some of the AloP peptides in *P. polymyxa*, especially the ones for which the fragment detected in the MS data is followed by a sequence longer than 6 aa, are further processed to yield shorter mature peptides (e.g., AloP4, P9, P15, and P16 in Fig 3B).

Our data suggest that the C-terminal 6 aa of AloP13 are processed and trigger transcriptional reprogramming similar to other peptides in the RRNPP family of quorum sensing systems. Whereas other RRNPP quorum sensing systems were further studied and characterized using synthetic constructs or deletion mutants of the receptors and peptides (21, 27, 41), *P. polymyxa* is a species challenging to genetically manipulate (46, 47). We overcame this challenge by cloning the Alo13 system in *B. subtilis* and demonstrating its activity in response to the addition of the mature peptide to the medium; this approach could be used to study other Alo systems in addition to Alo13. Furthermore, future development

of genetic systems for *P. polymyxa* ([48](ref)) may allow, in the future, deeper mechanistic studies of Alo systems in their native host.

We found that addition of AloP13 to growing cells results in immediate down-regulation of a set of genes, including those encoding the Spo0E-like phosphatase and additional Alo peptides. These immediate expression changes are expected to be translated into a signaling cascade that may involve alteration of the phosphorylation state of additional cellular components, ultimately giving rise to a functional phenotype. The phenotype controlled by AloP13, as well as those controlled by other AloP peptides, remains to be elucidated. Revealing the full network of functional phenotypes controlled by peptide communication in *P. polymyxa* may enable efficient harnessing of the plant growth–promoting traits of this organism for agricultural benefit in the future.

# Materials and Methods

## Identification of Alo systems in *P. polymyxa* ATCC 842

The aa sequences of all protein-coding genes of *P. polymyxa* N.R. Smith 1105, ATCC 842 were downloaded from IMG ([49](ref)) (on 31 January, 2017, IMG taxon ID: 2547132099, NCBI:txid 1036171) and scanned with TPRpred version 2.8 ([32](ref)). Genes with TPRpred scores of >75% probability of having TPR/pentatrico peptide repeats (PPRs)/SEL1 domains were further inspected. The genome environments of these genes were manually searched for downstream-annotated short ORFs with an N-terminal hydrophobic helix as predicted by the Phobius Web server ([50](ref)). For two TPR-containing genes, such downstream short ORFs were identified. BlastP ([51](ref)) was then used to search for homologs of these two genes among all *P. polymyxa* ATCC 842 proteins using an E-value cutoff of $1 \times 10^{-5}$. Homologs shorter than 250 aa were discarded. The remaining 16 genes were numbered *aloR1–aloR16*, and their cognate *aloP* genes were searched as above. In cases no cognate *aloP* gene could be identified, the last 100 bases of the *aloR* gene together with the 200 bases immediately downstream of the *aloR* were searched for short ORFs (30–50 aa) partially overlapping the 3′ of the *aloR* gene using Expasy translate tool ([52](ref)). N-terminal hydrophobic helices of these AloP peptides were predicted using the Phobius Web server ([50](ref)). Signal sequences were manually inferred in cases where the Phobius score was near the threshold based on homology to other AloP sequences, whose Phobius score was above the threshold.

## Phylogenetic analysis of Alo systems

The aa sequences of the 16 AloR proteins identified in *P. polymyxa* ATCC 842 were used as a query for iterative homology search against 38,167 bacterial and archaeal genomes downloaded from the IMG database ([49](ref)) on October 2017, using the "search" option in the MMseqs2 package ([53](ref)) (release 6-f5a1c) with "--no-preload –max-seqs 1000 –num-iterations 3" parameters. The hits were filtered with an E-value cutoff of $<1 \times 10^{-10}$. Homologs shorter than 250 aa or homologs found in scaffolds shorter than 2,500 nt were discarded.

For the analysis appearing in [Fig 2](ref), the 234 AloR homologs found in 14 *P. polymyxa* strains were aligned using the MAFFT multiple

sequence alignment server version 7 with default parameters ([54](ref)). The alignment was used to generate a maximum likelihood phylogenetic tree using IQ-TREE ([55](ref)) version 1.6.5 including ultrafast bootstrap analysis ([56](ref)) with 1,000 alignments ("-bb 1000") and using the "-m TEST" parameter to test for the best substitution model ([57](ref)) (best-fit model: JTT+F+R5 chosen according to BIC). The tree was visualized using the iTOL program ([58](ref)). A similar analysis with sequences of 11 *rap* genes from *B. subtilis* 168 used as an out-group produced the same tree structure and was used to define the tree root in [Fig 2A](ref). Clades were manually defined for AloR receptors based on the branching patterns in the phylogenetic tree ([Fig 2A](ref)), as well as similarity in the cognate AloP sequence and the immediate genomic environment. For [Fig 2B](ref), the dendrogram of the Alo receptors on the vertical axis of the matrix was generated by collapsing the tree shown in [Fig 2A](ref). The dendrogram of the phylogenetic relationship between the *P. polymyxa* strains shown on the horizontal axis of [Fig 2B](ref) was generated by multiple sequence alignment and IQ-TREE analysis of the conserved GyrA protein of all 14 *P. polymyxa* strains, including GyrA from *B. subtilis* as an out-group. For the data presented in Tables S1 and S3, AloR homologs were searched for their cognate *aloP* genes as described for *P. polymyxa* ATCC 842. For the data presented in [Fig S2](ref), NCBI accessions of representative RRNPP receptors were taken from Fig S5 of reference [59](ref) as follows: RapE ([AAM51168](ref)), RapA ([AAM51160](ref)), RapC ([AAT75294](ref)), NprR ([ABK83928](ref)), transcriptional regulator *Enterococcus faecalis* (T.reg *E. faecalis*, [NP_815038](ref)), DNA-binding protein *Bacillus anthracis* (DNAbd *B. anthracis*, [NP_843644](ref)), PlcR ([ZP_00739149](ref)), PrgX ([AAA65845](ref)), TraA [BAA11197](ref), transcriptional activator *L. monocytogenes* (T.act *L. monocytogenes*, [YP_013453](ref)), transcriptional regulator *L. casei* (T.reg_Lcas, [YP_805489](ref)), MutR ([AAD56141](ref)), and RggD ([AAG32546](ref)).

## Bacteria culture and growth conditions

*P. polymyxa* ATCC 842 was obtained from the BGSC (strain 25A2) and stored in −80°C as a glycerol stock. For all experiments, bacteria were first streaked on an lysogeny broth (LB) plate, grown in 30°C at least overnight, and colonies were used to inoculate round-bottom ventilated 15-ml tubes containing 4–5 ml LB medium, grown at 30°C with 200 rpm shaking.

## Preparation of growth media for MS experiments

To search for peptide fragments secreted to the growth medium, overnight 5 ml cultures of *P. polymyxa* or *B. subtilis* 168 were first washed by centrifuging for 5 min in 3,200*g* at room temperature and resuspended in 5 ml of chemically defined medium lacking any peptides described in [reference 60](ref). Washed cultures were then diluted 1:100 into 250 ml Erlenmeyer flasks with 50 ml defined media. The cells were grown in 30°C with 200 rpm shaking for 8 h for *P. polymyxa* log samples (OD ~0.1) and 24 h for the stationary samples (three biological replicates), and 5 h for *B. subtilis* log samples (OD ~0.3) and 8 h for the stationary samples (OD ~0.9, one biological replicate). At the designated time point, 23 ml samples were centrifuged for 5 min at 3,200*g* at 4°C, and two technical replicates of 10 ml of the supernatant for each sample were transferred to new tubes and flash-frozen immediately.

## Primers used in this study.

| Name | Sequence | Use |
|------|----------|-----|
| Spo0E_1_F | TGGCGAAAGAGGTGGGATTGA | RT-qPCR of *spo0E-like* gene (PPTDRAFT_04974)—primers set 1 |
| Spo0E_1_R | GGGTACGCTCTGTTGTCTACGA | RT-qPCR of *spo0E-like* gene (PPTDRAFT_04974)—primers set 1 |
| Spo0E_2_F | GAGTTGGATCGGCTCCTTAAT | RT-qPCR of *spo0E-like* gene (PPTDRAFT_04974)—primers set 2 |
| Spo0E_2_R | CTCGTGTTGCTTCTCTTGGA | RT-qPCR of *spo0E-like* gene (PPTDRAFT_04974)—primers set 2 |
| gyrB_F | GCCAGCGATACATTCCACTAT | RT-qPCR of *gyrB* gene for normalization |
| gyrB_R | CACGAGAGCCTTCGACATAAA | RT-qPCR of *gyrB* gene for normalization |
| Alo13_F | ATGGAAAATGCAACCACGATTC | Used to amplify the Alo13 locus for cloning (start of AloR13 ORF) |
| Alo13_R | ATCAATAAACCTCCTGTTCGGTG | Used to amplify the Alo13 locus for cloning (end of Spo0E promoter) |
| Alo13_Gib_F | CACCGAACAGGAGGTTTATTGATatgtcaaaaggagaagaactttttacag | Used for Gibson cloning of Alo13_F and Alo13_R amplified fragment (lower case letters correspond to beginning of sfGFP) |
| Alo13_Gib_R | GAACGAATCGTGGTTGCATTTTCCATgtttgtcctccttattagttaatcagc | Used for Gibson cloning of Alo13_F and Alo13_R amplified fragment (lower case letters correspond to RBS sequence following hyper-spank promoter) |
| AloP13_mut_F | TGTGCAAAGACGAAGAAGGTTTTTC | Used for site-directed mutagenesis of AloP13 start codon ATG > ACG |
| AloP13_mut_R | CCCCCTTAATAATAGATTTATAAATTTC | Used for site-directed mutagenesis of AloP13 start codon ATG > ACG |

## Sample preparation for MS analysis

A method based on protein interaction with a solid phase was applied to enrich for peptides present in the supernatants. Before usage, a disposable solid phase enrichment column (8B-S100-AAK; Phenomenex), packed with an 8.5-nm pore size, modified styrene-divinylbenzene resin was equilibrated by rinsing twice with 1 ml acetonitrile and twice with 1 ml water. Subsequently, 10 ml culture supernatant was loaded and unbound, potentially larger proteins were removed by washing twice with 1 ml water. Finally, the enriched sample fraction was eluted with 0.5 ml 70% acetonitrile (vol/vol) and evaporated to dryness in a vacuum centrifuge. Samples were resuspended in 0.1% (vol/vol) acetic acid before MS analysis.

## MS

The enriched peptides were loaded on an EASY-nLC 1000 system (Thermo Fisher Scientific) equipped with an in-house built 20-cm column (inner diameter 100 mm, outer diameter 360 mm) filled with ReproSil-Pur 120 C18-AQ reversed-phase material (3 mm particles, Dr. Maisch GmbH, Germany). Elution of peptides was executed with a nonlinear 86 min gradient from 1 to 99% solvent B (0.1% [vol/vol] acetic acid in acetonitrile) with a flow rate of 300 nl/min and injected online into a QExactive mass spectrometer (Thermo Fisher Scientific). The survey scan at a resolution of R = 70,000 and $3 \times 10^6$ automatic gain control target with activated lock mass correction was followed by selection of the 12 most abundant precursor ions for fragmentation. Data-dependent MS/MS scans were performed at a resolution of R = 17,500 and $1 \times 10^5$ automatic gain control target with a normalized collision energy of 27.5. Singly charged ions as well as ions without detected charge states or charge states higher than six were excluded from MS/MS analysis. Dynamic exclusion for 30 s was activated.

## Database search of MS results

Identification of peptides was carried out by database search using MaxQuant 1.6.3.4 with the implemented Andromeda algorithm ([61]) applying the following parameters: digestion mode unspecific; variable modification, methionine oxidation, and maximal number of five modifications per peptide; activated "match-between runs" feature. The false discovery rates of peptide spectrum match, peptide, and protein level were set to 0.01. Only unique peptides were used for identification. Two databases were used for peptide identification: a database for *P. polymyxa* ATCC 842 proteins downloaded from the IMG database ([49]) and supplemented with the sequences for the Alo systems (5,429 entries) and the reference proteome of *B. subtilis* 168 downloaded from the UniProt database (on 19 January, 2019) containing 4,264 entries. Common laboratory contaminations and reverse entries were added during MaxQuant search, and a peptide length of 5–35 aa was specified.

## Transcriptional response to peptides (RNA-seq experiments)

For RNA-seq experiments, synthetic lyophilized peptides were ordered from Peptide 2.0 Inc. and Genscript Corp at purity levels of 99% for the mature AloP13 (SHGRGG), 99.2% for the scrambled AloP13 (GRGSGH), and 81–93% for the AloP13 pro-peptide (SYFHNITIGNGSQITVSSHGRGG) and its scrambled version (VSGTRHGSFHSGIGNSGIYIQNT). The peptides were dissolved in DDW + 2% DMSO to a working stock concentration of 100 $\mu$M and kept in aliquots at −20°C. Overnight *P. polymyxa* cultures grown in LB were diluted 1:100 into 500-ml

Erlenmeyer flasks with 100 ml defined medium and grown at 30°C with 200 rpm shaking. At OD ~0.1, 45 ml of each culture was washed by centrifuging for 5 min in 3,200*g* at room temperature and resuspending the pellet in 45 ml chemically defined medium (60). The culture was split between 5-ml tubes containing each of the tested peptides at a final concentration of 5 $\mu$M, or a control tube containing a similar amount of DDW + 2% DMSO. The cultures were incubated in 30°C with 200 rpm shaking for 10 min, after which 1:10 cold stop solution (90% [vol/vol] ethanol and 10% [vol/vol] saturated phenol) was added to the samples. The samples were centrifuged for 5 min at 3,200*g* at 4°C, the supernatant was discarded, and the pellets were immediately flash-frozen and stored in 80°C until RNA extraction. The experiment was conducted three times on different days to produce independent biological replicates.

Frozen bacterial pellets were lysed using the Fastprep homogenizer (MP Biomedicals) and RNA was extracted with the FastRNA PRO blue kit (116025050; MP Biomedicals) according to the manufacturer's instructions. RNA levels and integrity were assessed using Qubit RNA HS Assay Kit (Q10210; Life Technologies) and TapeStation (5067-5576; Agilent), respectively. All RNA samples were treated with TURBO DNase (AM2238; Life Technologies). Ribosomal RNA depletion and RNA-seq libraries were prepared as described in reference 62, except that all reaction volumes were reduced by a factor of four.

RNA-seq libraries were sequenced using Illumina NextSeq platform, and sequenced reads were demultiplexed using Illumina bcl2fastq module. Reads were mapped to the reference genome of *P. polymyxa* ATCC 842 downloaded from IMG (IMG taxon ID: 2547132099, 65 contigs), as described in reference 62. RNA-seq–mapped reads were used to generate genome-wide RNA-seq coverage maps and reads-per-gene counts.

Raw counts of reads-per-gene for each of the three biological replicates were used as input for DESeq2 package analysis using R 3.6.0 (63), while accounting for batch effect (DESeqDataSet [dds] design model "design = ~batch + treatment"). Genes with normalized mean count <100 were discarded, and a significant adjusted *P*-value (false discovery rate < 0.05) and fold change >2 or <0.5 were considered as the threshold for differentially expressed genes in each contrast (mature versus control and scrambled versus control). The volcano plot in Fig 4A was plotted using EnhancedVolcano R package (64), based on the results of the DESeqs2 analysis.

### RT-qPCR analysis

Bacteria were treated with the AloP13 mature peptide, scrambled mature peptide, and no peptide as described for the RNA-seq experiments, and cell pellets were collected from two biological replicates. Total RNA was extracted from the cells as described above, and 1 $\mu$g of DNase-treated RNA was then reverse-transcribed using high-capacity cDNA reverse-transcription kit with random primers according to the manufacturer's instructions (Applied Biosystems). Quantitative PCR was performed with SYBR Green PCR Master Mix (Applied Biosystems/Thermo Fisher Scientific) and gene-specific primers listed below in triplicates, on Viia7 platform (Applied Biosystems). Error bars indicate SD of 2–3 measurements for each sample, and the housekeeping gene *gyrB* was used for normalization.

### Cloning of the Alo13 locus in *B. subtilis*

The Alo13 locus including the *aloR13* ORF, *aloP13* ORF, and the intergenic region until the *spo0E-like* ORF was amplified using primers Alo13_F and Alo13_R (listed below). The fragment was cloned using NEBuilder HiFi DNA Assembly Master Mix (New England Biolabs) with primers Alo13_Gib_F and Alo13_Gib_R under an IPTG-inducible *hyper-spank* promoter with a super-folder GFP (sfGFP) into an AmyE integration cassette with ampicillin and spectinomycin resistance and transformed into chemically competent *Escherichia coli* DH5$\alpha$. The plasmids were then extracted and transformed into *B. subtilis* BEST7003 as described before (65). To introduce a point mutation in the start codon of AloP13, primers AloP13_mut_F and AloP13_mut_R were used with Q5 Site-Directed Mutagenesis Kit (New England Biolabs) according to the manufacturer's instructions. The integrity of the sequence was verified by Sanger sequencing.

### Fluorescence assay

Overnight LB cultures of *B. subtilis* BEST7003 with the Alo13-GFP integrated construct were diluted 1:100 into defined medium with the synthesized peptides at various concentrations, and with 0 or 100 $\mu$M IPTG (Isopropyl $\beta$-D-1-thiogalactopyranoside) as an inducer. Optical density and fluorescence measurements were performed in triplicates using a TECAN Infinite 200 plate reader in a 96-well plate.

## Data Availability

All MS data (Fig 3 and Table S4) have been deposited to the ProteomeXchange Consortium via the PRIDE (66) partner repository with the dataset identifier PXD015319. All raw RNA-seq datasets (Fig 4 and Tables S5 and S6) were deposited in the European Nucleotide Database (ENA), study accession no. PRJEB34369.

## Supplementary Information

## Acknowledgements

We thank Yael Helman, Yaara Oppenheimer-Shaanan, Ohad Herches, Alon Savidor, Etai Rotem, Gilad Yaakov, Tabitha Bucher, Maya Schuldiner, Nofar Mor, Daniel Zeigler, and Jürgen Bartel for their expertise and technical assistance. We thank the Sorek lab members for fruitful discussions and helpful suggestions. R Sorek was supported, in part, by the Israel Science Foundation (personal grant 1360/16), the European Research Council (ERC) (grant ERC-CoG 681203), the German Research Council (DFG) priority program SPP 2002 (grant SO 1611/1-1), and the Knell Family Center for Microbiology. M Voichek is a Clore Scholar and was supported by the Clore Israel Foundation. S Maaß and D Becher were supported by the German Research Council (DFG) priority program SPP 2002 (grant BE 3869/5-1).

## Author Contributions

M Voichek: conceptualization, data curation, software, formal analysis, validation, investigation, visualization, methodology, and writing—original draft, review, and editing.

S Maaß: software, formal analysis, validation, investigation, and methodology.

T Kroniger: methodology.

D Becher: software, supervision, funding acquisition, validation, investigation, and methodology.

R Sorek: conceptualization, resources, supervision, funding acquisition, investigation, visualization, methodology, project administration, and writing—original draft, review, and editing.

## Conflict of Interest Statement

The authors declare that they have no conflict of interest.

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
