## [Reviewer comments · Life Science Alliance]

Life Science Alliance

Peptide-based quorum sensing systems in *Paenibacillus polymyxa*

Maya Voichek, Sandra Maaß, Tobias Kroniger, Doerte Becher, and Rotem Sorek
DOI: <https://doi.org/10.26508/lsa.202000847>

Corresponding author(s): Rotem Sorek, The Weizmann Institute of Science

Review Timeline:	Submission Date:	2020-07-13
	Editorial Decision:	2020-07-21
	Revision Received:	2020-07-27
	Accepted:	2020-07-28

Transaction Report:

Please note that the manuscript was previously reviewed at another journal and the reports were taken into account in the decision-making process at Life Science Alliance.

Reviewer #1 Review

Report for Author:

This manuscript reports the identification for the first time of a quorum sensing system in the Paenibacillaceae. The quorum sensing system identified encodes a Phr-type peptide as the quorum sensing molecule and a RRNPP protein as the peptide receptor. The researchers identified these systems through a straightforward and fruitful approach, searching for genes with TPR domains, as found in the RRNPP proteins, screening for adjacent, small open reading frames that encoded a protein with a signal sequence for secretion. Candidate gene cassettes were then used to identify orthologs within the Paenibacillaceae. The remarkable finding in this study is the large number of quorum sensing gene cassettes identified, up to 27 cassettes in one genome. While the Phr-type peptide signaling systems were known to be highly enriched in the Bacilli, the number of cassettes in Paenibacillaceae are 2-3-fold higher. The researchers name the peptide genes *aloP* and the receptor genes, *aloR*. The researcher carried out a number of studies that convincingly indicated that the *AloP* protein is secreted and that the *AloR* protein regulates gene expression in response to an *AloP* peptide.

This manuscript is well written and there is sufficient data to support the conclusions. The significance of this paper is in identifying for the first-time quorum-sensing systems for Paenibacilli

and identifying the largest number of quorum sensing systems in one bacterium to date. This would seem to me to be of importance to those studying quorum sensing, but not of more broad interest. There is not some new fundamental discovery about quorum sensing.

Reviewer #2 Review

Report for Author:

Voichek et al describe a sequence domain-based search aimed at finding peptide-based QS systems in Paenibacilli. Their approach seems to have identified several putative receptor-signal gene pairs which are quite diverged from previously known systems, but still appear to have the hallmarks of the systems being sought. A broader search revealed these putative systems to be widespread in related species, and easily grouped. Adding synthetic peptide from one such system elicited a transcriptional response, as expected, and further work showed this to be dose dependent.

Overall, I think the paper is well written and adds some useful information to the field. I have no major issues. I do, however, feel that considerable experimental work could be done to strengthen the paper and increase its value to the community. A novel contribution could come with functional verification of more systems and analysis of the interplay among them.

Ln 102-114. I think this section should make it clear that these conserved sequences in the various *P. polymyxa* strains are predicted or putative Alo systems, since there is no functional test.

Ln 145-148. The presence of these systems in Paenibacilli is indeed interesting and suggestive. However, I suggest that while they remain mostly untested, and while sequence space remains reasonably poorly explored in general, the authors refrain from conclusions like Paenibacilli are among the most "communicative" bacteria studied to date.

Ln 171-180. I agree with the likely situation that detection of the peptide that lacks the C-terminus probably indicates cleavage and release of that C-terminus. However, if possible I think it would be good to show that the full-length peptide exists somewhere (maybe along with the signal sequence prior to secretion). Otherwise, the mature signal peptide is inferred but not detected either alone or as part of the pre-processed peptide.

Ln 191-206. I like this experiment. I wonder, was the synthetic peptide treatment and RNA-seq attempted on a strain with a deletion (or some more focused mutation) of the predicted receptor protein? Also, an important experiment is RNA-seq in a strain carrying a deletion of the signal peptide gene. Does the gene expression pattern in the mutant fail to change compared to parental strain as the population increases? Is there a phenotype that could be predicted to be associated with regulation by this signal, which could be evaluated in a mutant?

Ln 248-255. It is interesting to see such a skew of affected genes toward other peptide signals. Definitely something to follow up in a new paper. Here though I'm curious about whether you'd expect to see altered expression of cognate receptors too, since the genes look closely linked, and are maybe operons.

Ln 271-273. Because all but one of the systems identified were only examined by sequence analysis (and not functional tests) I suggest rephrasing to something like: "In this work we identified

a large family of putative peptide-based quorum sensing systems in Paenibacillaceae bacteria, and functionally characterized one of these systems in the type strain *P. polymyxa* ATCC 842."

Ln 283-284. I would probably argue against the statement about having the largest number of QS systems to date. For a start, only one of them is functionally tested. And more importantly, this kind of claim adds nothing significant to the science of this paper and is likely to become out of date. The large number of systems is interesting in terms of the complexity, as mentioned in the preceding sentences, but whether there are more or less than in other bacteria is uninteresting. It's not a competition!

Fig 1b. Maybe, if it's known, the peptide cleavage site that releases the mature signal peptide could be shown.

Fig 3. I think it would help to include the explanation that the C-terminal sequence thought to be the mature peptide was not detected in most cases, and that its presence was inferred by the existence of peptides that lack it (and are presumably processed).

Fig 4f. I get the data, but find the presentation of the information to be a bit difficult to look at. The color key on the right side is not easy to reconcile with all parts of the graph. E.g. the bright green for high level expression doesn't seem to have a corresponding entry in the key, and similarly the reds and oranges of low expression don't seem to have matches in the key either.

Reviewer #3 Review

Report for Author:

The manuscript by Voichek et al. reports on the discovery of RRNPP-type quorum sensing systems in *Paenibacillus polymyxa*. RRNPP-type systems were discovered more than 40 years ago in *Bacillus subtilis* and are known to accumulate in the genomes of many Bacilli. Related systems have been discovered in a number of other G+ bacteria including Enterococci and Streptococci. In general, molecular studies in Paenibacilli are still very much in their infancy. Since this is an important class of bacteria due to their potential for biotechnological applications work on *Paenibacillus* is expected to gain increasing attention.

Voichek et al. support their discovery by bioinformatics and experimental data. Candidate systems were identified using available *Paenibacillus* genomes. A sample search in *P. polymyxa* exploited homology of receptors searching for characteristic TRP-motifs of RRNPP-type receptors and the fact that the signaling peptide is typically derived from a small ORF found in immediate proximity to the receptor gene. Additional systems were then identified using homology to the first candidate receptors including other bacteria.

In *Paenibacillus polymyxa* cleavage products of the peptides are detected in culture supernatant. This also provides indirect evidence for the production of the mature signaling peptides, although these evade current mass spec. The authors use global gene expression analysis to identify genes that are controlled by the addition of a synthetic (putative) signaling peptide. One very likely direct target gene is a Spo0E-like phosphatase. This is supported by q-PCR and fluorescent gene reporter output by transplanting the signaling system with its putative target into a heterologous *B. subtilis* host. Perhaps noteworthy the authors also find that the addition of the signaling peptide affects the expression of other RRNPP-type signaling systems in *P. polymyxa*, either directly or

indirectly. This indicates cross-regulation, a phenomenon also seen in other G+ bacteria.

This study reports on important and sound results, which should be of high interest for the small but growing Paenibacillus field. The findings will likely kick-start more research on cell-cell communication. The study should also be of interest to G+ cell-cell communication field as it further expands the range of bacteria in which RRNPP-type signaling system function.

However, I have some difficulties in seeing the impact of this work for the molecular systems biology field. Specifically there seems to be little conceptual or technical advance compared to previous studies in other organisms. I am therefore a bit surprised that the authors have not considered a microbiology journal for communicating their findings.

Reviewer #4 Review

Report for Author:

This is a very interesting manuscript with solid data analysis.

The manuscript was well prepared and organized.

I noticed the full manuscript was already preprinted by bioRxiv in year 2019:

<https://www.biorxiv.org/content/biorxiv/early/2019/09/12/767517.full.pdf>

In addition, this manuscript was also posted in Researchgate and public available for download:

[https://www.researchgate.net/publication/335792658_Peptide-](https://www.researchgate.net/publication/335792658_Peptide-based_quorum_sensing_systems_in_Paenibacillus_polymyxa)

[based_quorum_sensing_systems_in_Paenibacillus_polymyxa](https://www.researchgate.net/publication/335792658_Peptide-based_quorum_sensing_systems_in_Paenibacillus_polymyxa) since Sept 2019: Peptide-based quorum sensing systems in Paenibacillus polymyxa

September 2019

DOI: 10.1101/767517

LicenseCC BY-NC-ND 4.0

July 21, 2020

RE: Life Science Alliance Manuscript #LSA-2020-00847-T

Prof. Rotem Sorek
The Weizmann Institute of Science
Department of Molecular Genetics
Meyer bldg 210A
Rehovot 76100
Israel

Dear Dr. Sorek,

Thank you for submitting your revised manuscript entitled "Peptide-based quorum sensing systems in *Paenibacillus polymyxa*". The manuscript was assessed by expert reviewers at another journal, and the editors transferred those reports to us with your permission. We would be happy to publish your paper in Life Science Alliance pending final revisions necessary to meet our formatting guidelines.

- please make sure the author names in the manuscript match the author names in our system
- please upload your main figures and your supplementary figures as single files
- please add a figure legend section for the main and supplementary figures to your manuscript text
- please upload your manuscript as an editable doc file
- please upload your tables as editable doc or excel files
- please list 10 authors et al. in the references

A. FINAL FILES:

B. MANUSCRIPT ORGANIZATION AND FORMATTING:

Sincerely,

Reilly Lorenz
Editorial Office Life Science Alliance
Meyerhofstr. 1
69117 Heidelberg, Germany
t +49 6221 8891 414
e contact@life-science-alliance.org
www.life-science-alliance.org

July 28, 2020

RE: Life Science Alliance Manuscript #LSA-2020-00847-TR

Prof. Rotem Sorek
The Weizmann Institute of Science
Department of Molecular Genetics
Meyer bldg 210A
Rehovot 76100
Israel

Dear Dr. Sorek,

Thank you for submitting your Research Article entitled "Peptide-based quorum sensing systems in *Paenibacillus polymyxa*". It is a pleasure to let you know that your manuscript is now accepted for publication in Life Science Alliance. Congratulations on this interesting work.

DISTRIBUTION OF MATERIALS:

Again, congratulations on a very nice paper. I hope you found the review process to be constructive and are pleased with how the manuscript was handled editorially. We look forward to future exciting submissions from your lab.

Sincerely,

Reilly Lorenz
Editorial Office Life Science Alliance
Meyerhofstr. 1
69117 Heidelberg, Germany
t +49 6221 8891 414
e contact@life-science-alliance.org
www.life-science-alliance.org